# Cortical Thickness of the Orbitofrontal Cortex in Patients with Alcohol Use Disorder

**DOI:** 10.3390/brainsci13040552

**Published:** 2023-03-25

**Authors:** Murad Atmaca, Muhammed Fatih Tabara, Mustafa Koc, Mehmet Gurkan Gurok, Sema Baykara, Sevda Korkmaz, Osman Mermi

**Affiliations:** 1Department of Psychiatry, School of Medicine, Firat University, Elazig 23119, Turkey; 2Department of Psychiatry, Bingol State Hospital, Bingol 12000, Turkey; 3Department of Radiology, School of Medicine, Firat University, Elazig 23119, Turkey

**Keywords:** alcohol use disorder, orbitofrontal cortex (OFC), cortical thickness

## Abstract

Aims: In the present study, it was hypothesised that compared to healthy control subjects, significant differences in the cortical thickness of the orbitofrontal cortex (OFC) region of the brain, which is relevant to both impulsivity and decision making, would be identified. Methods: The subject groups included in the study were composed of 15 individuals who met the criteria for alcohol use disorder, according to the Diagnostic and Statistical Manual of Mental Disorders Fifth Edition (DSM 5) diagnostic criteria based on the Structured Clinical Interview for DSM 5 (SCID), and were admitted to the Firat University School of Medicine Department of Psychiatry or were hospitalised, and 17 healthy control comparisons were made. The volumes of and cortical thickness of the OFC were measured in the subjects. Results: It was found that patients with alcohol use disorder had reduced volumes of the OFC bilaterally and a thinner cortical thickness of the same region bilaterally compared to those of the healthy control comparisons. Conclusions: Consequently, it is suggested that the OFC region of the brain appears to be statistically significantly smaller in patients with alcohol use disorder, both in terms of cortical thickness and volume, compared to healthy controls. Future research should focus on the status of these relationships longitudinally and should assess the causality of the association with the treatment response.

## 1. Introduction

Alcohol use disorder is both a psychiatric disorder and an important public health problem, which is very common in the world and manifests itself with complications. The Diagnostic and Statistical Manual of Mental Disorders, version 5 (DSM 5) establishes the diagnosis of alcohol use disorder when at least two of the eleven basic symptoms are met in a period of at least 12 months. The most well-known clinical feature is frequent and heavier-than-planned drinking and intense but unsuccessful efforts to quit alcohol [1]. It is a very common psychiatric disorder that affects a significant portion of the world’s population [2]. It has been reported that one third of the male population and one quarter of the female population have met the criteria for alcohol use disorder at least once in their life in the United States of America [3,4]. This result demonstrates the importance and burden of alcohol use disorder in the world. Meanwhile, the DSM 5, different from previous editions, deals with alcohol use disorder as a spectrum, not as a present or absent condition, and categorises the disorder from mild to severe [1]. The exact cause of how alcohol use disorder manifests in patients is unknown, and it is probably due to the mixed effects of neurobiological factors and psychosocial factors, as in other psychiatric disorders. Although it is difficult to summarise the neurobiological etiopathogenesis, there are probably many factors, from hormone levels to neurochemical changes, from genetic factors to structural and functional alterations in different areas of the brain. Important findings were obtained in structural brain studies, which are also the subject of our current study, in patients with alcohol use disorder [5]. The results of the first studies on this subject revealed insignificant volumetric reductions in the hippocampus, thalamus and mammillary bodies [6,7,8,9,10]. Although there are studies reporting the opposite, it has also been shown that hippocampal volume deficits are associated with age in patients with alcohol use disorder [9,11,12,13]. Lateral ventricular enlargement, which can be observed in patients with chronic psychiatric disorders, was detected in a group of patients with chronic alcohol use disorder who started using again after a break [14,15]. On the other hand, when looking at the structural studies of the basal ganglia, significant volumetric reductions were determined, especially in the putamen, nucleus caudatus and nucleus accumbens [16,17]. When we come to the forebrain regions, in general, especially in elderly patients with alcohol use disorder, it can be seen that significant volume deficits have been defined. Again, when compared to the posterior regions, corpus callosum volume reductions, which are more pronounced in the anterior regions, were also determined as another important finding. An important finding came from patients with alcohol use disorder who stopped drinking: a period of abstinence has been shown to reverse both grey and white matter changes in different brain areas and leads to some improvement [18,19,20,21]. Our study team also performed some structural brain studies in patients with alcohol use disorder. In one of them, the pineal gland volumes in fifteen patients with alcohol use disorder were compared with those of healthy control subjects by using magnetic resonance imaging (Atmaca et al., unpublished study). In that study, the team detected that the mean volume of the pineal gland was significantly smaller in patients with alcohol use disorder compared to healthy controls (86.02 ± 5.11 mm^3^ vs. 107.35 ± 30.04 mm^3^, with a statistically significant difference of *p* < 0.01). In addition, since gender, age, and total brain volumes could affect absolute pineal gland volumes, these variables in the general linear model were controlled and gland volumes in patients with alcohol use disorder and healthy controls and compared. After these controls, it was determined that smaller pineal gland volumes persisted statistically in patients with alcohol use disorder, concluding that patients with alcohol use disorder had smaller pineal gland volumes compared to those of the healthy control subjects. In another study, it was aimed to determine whether there is a difference in pituitary gland volumes in patients with alcohol use disorder compared to healthy people. In this study, it was determined that absolute pituitary gland volumes were significantly smaller in patients with alcohol use disorder compared to healthy controls (58.02 ± 7.24 mm^3^ in patients with alcohol use disorder vs. 83.08 ± 12.11 mm^3^, with a statistically significant difference of *p* < 0.01) (Atmaca et al., unpublished study). Finally, studies on the orbitofrontal cortex, which is the region where impulsivity is modulated and which is highly likely to be associated with alcohol use disorder, revealed that restarting alcohol in patients with alcohol use disorder who have been sober for a while may be associated with bilaterally decreased orbitofrontal cortex volumes [22,23,24].

Recently, determining cortical thicknesses in various psychiatric disorders, especially to predict treatment response, has been an important area of interest in neuroimaging. In this context, the work of Kühn et al. is a prime example. In this study, researchers determined significant cortical thinning in the dorsal and subgenual anterior cingulate cortex (ACC) compared to healthy control subjects, and they determined that there was a significant correlation between left dorsal ACC thinning and the Yale–Brown Obsession-Compulsion Scale obsession subscore [25]. Bae et al. reported significantly thinner cortical thickness in patients with alcohol dependence than healthy comparison subjects in the left superior frontal cortical region while Dai et al. demonstrated significant volumetric decreases in cortical thickness in the bilateral temporal, insular, precentral, and dorsolateral prefrontal gyri of patients with alcohol use disorder [26,27]. Likewise, cortical thickness alterations were supported by another investigation for both the left and right hemispheres [28]. Venkatasubramanian et al. examined cortical volume, thickness, surface area and the local gyrification index (LGI) by using a completely automated surface-based morphometric analysis and reported that Y-BOCS obsession scores were significantly negatively correlated with the left frontal pole volume whereas Y-BOCS compulsion scores were significantly negatively correlated with the right ACC volume and surface area and right lateral OFC LGI [29]. Hoexter et al. also examined whether OFC cortical thickness could be an important indicator for predicting treatment response in patients with OCD who were treatment-naive, and ultimately revealed that left and right medial OFC cortical thickness is the most important indicator for predicting treatment response [30]. There are two indirect studies on alcohol or substance use disorder. In the first of these, Henderson et al. compared healthy adolescents with and without a family history of alcohol use disorder, and determined a significantly thinner cortical surface in frontal and parietal areas, especially in the medial and lateral orbitofrontal and superior parietal cortices, in individuals with a family history compared to those without, considering that frontal and parietal structural differences in family history positive individuals that might underlie cognitive and behavioural characteristics related to AUD risk [31]. In association with this study, Hill et al. carried out a study in which volumes of the OFC were examined in a group of adolescents and emerging adults with or without of family history of alcohol use disorder and found that there were no volumetric differences between groups, whereas the proportion of the right to left OFC volume was statistically significantly lower in family history positive subjects compared to those of negative ones [32]. On the other hand, Harper et al. examined whether relationships between alcohol, cannabis and tobacco use disorders and OFC thickness reflected the potential causal effects of familial risk or substance use disorders related consequences [33]. While the researchers determined that the thickness of the medial orbitofrontal cortex was significantly thinner in the study group than in the healthy controls, they determined that there was no such difference for the lateral orbitofrontal cortex, suggesting that the experience of an alcohol or cannabis use disorder in the emerging adulthood period, seems to reduce the thickness of the medial orbitofrontal cortex, considering that this is a region related to value-guided decision making.

All the aforementioned knowledge prompted the evaluation of the orbitofrontal cortex volumes and thickness in patients with adult alcohol use disorder, and it was hypothesised that compared to healthy controls, significant differences in cortical thickness of this region would be obtained, which would be relevant to both impulsivity and decision making.

## 2. Methods

### 2.1. Subjects

In this study, fifteen patients (18–65 years old) who applied to the Psychiatry Department of Firat University, Faculty of Medicine and had alcohol use disorder, according to the Diagnostic and Statistical Manual of Mental Disorders, version IV-text revised (DSM-IV-TR) and seventeen healthy control subjects were taken as the study group [34]. All patients and control subjects were informed in detail about how the study would be conducted and were included in the study on a voluntary basis. It is important to note that the subjects of the present study were those of another unpublished investigation in which pineal gland volumes were examined (Atmaca et al., unpublished study). Written informed consent was obtained from all patients and healthy control subjects confirming that they wanted to voluntarily participate in the study. The present study was approved by the Firat University School of Medicine Local Ethics Committee, and the principles reported in the Declaration of Helsinki were strictly followed in all study procedures. The Turkish version of the Structured Clinical Interview for DSM-IV (SCID) according to the Diagnostic and Statistical Manual of Mental Disorders Fourth Version (DSM-IV) was used to establish the diagnosis of alcohol use disorder for all patients [35]. All patients who participated in the study were right-handed. Interviews with patients and their first-degree relatives were held in a place reserved for this investigation in the clinic. Some exclusion criteria were used to exclude patients who were not wanted to be included in the study. These were: the existence of comorbid Axis I psychiatric disorder other than depression was not included since depression is a frequent comorbid disorder, presence of a serious medical illness, a history of serious head trauma, a presence of a congenital malformation in the brain, presence of any contraindication to MRI examinations, such as a cardiac stent or joint device, an unstable dose of psychotropic drug use at least for two months. All of the patients were in a period of abstinence, with a mean duration of 4.40 ± 3.18 months. Healthy control subjects were selected among the well-being population who had been invited to obtain their magnetic resonance imaging for our study. Some of them were from the same hospital, working as hospital staff. Similar exclusion criteria were used to exclude healthy control subjects. They were: the presence of any Axis I psychiatric disorder, the presence of any severe medical problems, history of a severe head trauma, the existence of congenital malformation in the brain, and any contraindication to MRI investigations, such as the existence of a cardiac stent or joint device.

The Michigan Alcoholism Screening Test (MAST), Severity of Alcohol Dependence Questionnaire (SADQ-C), and the Alcohol Use Disorders Identification Test (AUDIT) were used to identify the level and status of the use of alcohol in the patients [36,37,38].

### 2.2. Magnetic Resonance Imaging (MRI) Procedure

Necessary psychological tests were applied to all patients and control subjects in the section reserved for clinical work. Information was given about the design of the MRI study. They were informed about what they would experience in the MRI procedure and the duration of the procedure. In addition, it was stated that an anxiolytic drug could be given to patients and controls if they so wished. All magnetic resonance imaging was performed in the Firat University School of Medicine, Department of Radiology, Neuroradiology section. All images were performed with a 3.0 Tesla scanner in a position with the head comfortably held (Philips Ingenia, Philips Medical Systems, Best, The Netherlands). The whole brain was scanned with a 3-D fast field echo (FFE) T1-weighted dataset. T1-weighted images were obtained in the axial, sagittal, and coronal plane in 1 mm contiguous sections. To obtain the images, some parameters were utilised: TR was 8 ms, TE was 3.7 ms, and the flip angle was 8°, with a 240 × 222 mm matrix. In addition, magnetic field uniformity was provided before every scan. Before the thicknesses were measured by an experienced radiologist (M.K.), the presence of significant structural abnormalities of patients with alcohol use disorder and healthy control subjects were roughly revised. However, no considerable structural abnormalities were determined for the subjects included in the study.

Image analyses were carried out using the FreeSurfer program (version 4.4), a free program to use and available to download online, which was used to reconstruct the cortical surfaces and detect the cortical thickness from magnetic resonance images (http://surfer.nmr.mgh.harvard.edu, accessed on 10 November 2022). For all processes, the method of Epstein et al. was followed [39]. The process utilised to analyse each subject’s brain images is available on the FreeSurfer website. Scans were processed by a radiologist who was blind to both the diagnosis of patients with alcohol use disorder and to whom was a patient or a healthy control subject. The process included motion correction, extraction of the brain tissue, transformation to Talairach space, segmentation and parcellation into the OFC region as described by Desikan et al. [40]. After this process, scans were also examined by a senior radiologist (M.K.) to ensure that segmentation and parcellation were performed correctly. In addition, scans were also reviewed for the presence of any artifact. However, no artifact was found. Uncorrected OFC cortical volumes (mm^3^) were divided by the total segmented brain volume (mm^3^) and then multiplied by 1000, to obtain the corrected cortical volumes. In contrast to Epstein et al.’s study, the OFC region was investigated in two parts, as left and right OFC [39]. Assistance was obtained from standard neuroanatomy atlases during all measurements [41,42,43]. On the other hand, the boundaries of the OFC region were adapted from Portas et al. and Riffkin et al. [44,45]. By using those guidelines, the same procedure as in our previous studies was preferred [46,47]. When determining the volumes of the OFC, the superior boundary of the OFC was defined by a line extending from the anterior commissure to the posterior commissure. The point at which the olfactory sulcus first appeared was determined as the posterior landmark. As expected, while the inferior boundary of the OFC was selected the most inferior aspect of the cortex, the most lateral edge of the brain cortex was also accepted as the lateral boundary of OFC. Finally, the longitudinal fissure was accepted as the medial boundary of the OFC. Sample imaging owing to the OFC is presented in Figure 1, whereas the images obtaining from the Freesurfer program are presented in Figure 2.

### 2.3. Statistical Analysis

The Statistical Package for the Social Sciences, version 22.0 (SPSS) (SPSS Inc., Chicago, IL, USA) was used to perform statistical analyses. The alpha value was accepted as 0.05 for statistical significance. The independent t test was used when performing cortical thickness and volumetric comparisons. For categorical data, the chi-square test was preferred. On the other hand, in order to eliminate the effect of age, gender and total brain volumes, comparisons between patients with alcohol use disorder and healthy control subjects were made using the general linear model in the SPSS, version 22.0. Finally, Spearman’s correlation test was used for correlation analyses between volumetric and thickness data and clinical and sociodemographic data.

## 3. Results

Firstly, since an attempt had been made to match the two groups, there was no difference between patients with alcohol use disorder and healthy controls in terms of age, gender distribution, education, and handedness. All patients and healthy control subjects participating in the study were right-handed. As can be seen in Table 1, no statistically significant difference was found between patients with alcohol use disorder and healthy controls in terms of age (the mean age ± SD = 32.52 ± 5.90 years old for healthy control subjects, and mean age ± SD = 36.86 ± 11.03 years for patients with an alcohol use disorder) (*p* > 0.05). The scale scores of the MAST, SADQ-C, and AUDIT owing to alcohol use disorder are given in Table 1. All sociodemographic and clinical data are also shown in Table 1.

When looking at the volumetric values of the brain, no significant difference between patients with alcohol use disorder and healthy controls in terms of total brain volume, brain total grey matter and total white matter volumes (*p* > 0.05) was found. However, when looking at OFC volumes, significant differences were determined. It was detected that patients with alcohol use disorder had significantly smaller mean left side volumes compared to those of healthy control subjects, with a mean volume of 12.53 cm^3^ in patients with alcohol use disorder (SD ± 1.12) compared to 14.38 cm^3^ in the healthy control subjects (SD ± 1.13), demonstrating that patients with alcohol use disorder had statistically significantly smaller volumes for the left OFC (t = −4.50; *p* < 0.001). As for the right OFC volumes, smaller right OFC volumes of patients with alcohol use disorder compared to those of healthy control subjects, with a mean volume of 12.07 cm^3^ in patients with antisocial personality disorder (SD ± 0.87) compared to 14.07 cm^3^ in the healthy control subjects (SD ± 1.09), were found demonstrating that patients with alcohol use disorder had statistically significantly reduced volumes for the right OFC (t = −5.57; *p* < 0.001). On the other hand, since age, gender and total brain volumes can affect OFC volumes and play a confounding role, The General Linear Model was used in SPSS to control them. In this context, it was determined that the patient group had statistically significantly reduced left and right OFC volumes compared with healthy controls after controlling for age, sex, and whole-brain volume (for the left side, *F* = 8.34, *p* < 0.001 for gender distribution, *F* = 10.12, *p* < 0.001 for age, and *F* = 7.88, *p* < 0.001 for whole-brain volume; for the right side, *F* = 10.56, *p* < 0.001 for gender distribution, *F* = 9.09, *p* < 0.001 for age, and *F* = 11.28, *p* < 0.001 for whole-brain volume).

As for the cortical thickness analyses, for the left medial and lateral OFC, it was determined that the mean cortical thicknesses were statistically significantly reduced in patients with alcohol use disorder compared to that of healthy control subjects (for the left medial OFC, *t* = −3.72, *p* < 0.01; for the left lateral OFC, *t* = −3.22, *p* < 0.01). Likewise, when computing the right medial and lateral OFC region analysis to determine differences in the cortical thickness between patients with alcohol use disorder and healthy control subjects, it was found that a significant reduction in cortical thicknesses of that region in the patient group compared to those of healthy comparisons existed (for the right medial OFC, *t* = −4.11, *p* < 0.001; for the right lateral OFC, *t* = −4.67, *p* < 0.001).

The error bars for OFC volumes and cortical thicknesses in groups are given in Figure 3 and Figure 4.

To examine the relationships between volumetric and thickness data of the OFC, and sociodemographic characteristics such as age and sex, clinical characteristics such as disease duration, and scale scores, the Spearman’s correlation test was used. It was detected that there were no correlations between the volumetric and thickness data of the OFC any demographic and clinical variables and scale scores both in patients with alcohol use disorder and the healthy control comparisons (*p* > 0.05).

## 4. Discussion

In this study, OFC cortical thickness was determined in patients with alcohol use disorder. Before starting the discussion, it is necessary to emphasise some important findings, as this study is the first study on this subject in the literature, to the best of our knowledge. It was detected that patients with alcohol use disorder had significantly smaller mean left- and right-side volumes compared to that of healthy control subjects. As for to cortical thickness analysis, for both sides of the OFC, it was determined that the mean cortical thicknesses were statistically significantly reduced in patients with alcohol use disorder compared to those of healthy control subjects.

As mentioned in the introduction, studies on the OFC, which is the region where impulsivity is modulated and which is highly probably to be associated with alcohol use disorder, revealed that restarting alcohol in patients with alcohol use disorder who have been sober for a while may be associated with bilaterally decreased OFC volumes [22,23,24]. In his review, Moorman emphasised that the OFC had an important role in the controlling of flexible, goal-directed behaviour, as well as its relationship with reward identification and acquisition, it is likely that the OFC would emerge as a key region in regulating excessive alcohol seeking seen in alcohol use disorder [48]. In this context, it is necessary to focus on the significant OFC deficit that was determined despite the low sample number. Of course, it is also very important to determine whether this has a functional equivalent. For this, functional studies with large samples are needed. Recently, determining cortical thicknesses in various psychiatric disorders, especially to predict treatment response, has been an important area of interest in neuroimaging [30]. In this context, Hoexter et al. also examined whether OFC cortical thickness could be an important indicator for predicting treatment response in patients with OCD who were treatment-naive, and ultimately revealed that left and right medial OFC cortical thickness is the most important indicator for predicting treatment response [30]. There are two indirect studies on alcohol or substance use disorder. In the first of these, Henderson et al. compared healthy adolescents with and without a family history of alcohol use disorder, and found a significantly thinner cortical surface in frontal and parietal areas, especially in the medial and lateral orbitofrontal and superior parietal cortices, in individuals with a family history compared to those without, considering frontal and parietal structural differences in family-history-positive individuals that might underlie cognitive and behavioural characteristics related to AUD risk [31]. In association with that investigation, Hill et al. carried out an investigation in which volumes of the OFC were examined in a group of adolescents and emerging adults with or without a family history of alcohol use disorder, and detected that there were no volumetric differences between groups, whereas the proportion of the right to left OFC volume was statistically significantly lower in family-history-positive subjects compared to those who had a negative family history of alcohol use disorder [32]. In addition, Harper et al. investigated whether relationships between alcohol, cannabis and tobacco use disorders and OFC thickness reflected the potential causal effects of familial risk or substance-use-disorders-related consequences [33]. The investigators identified that the thickness of the medial OFC was significantly thinner in the study subjects compared to that of the healthy control subjects; they did not determine such difference for the lateral OFC, suggesting that an alcohol or cannabis use disorder in the emerging adulthood period seems to reduce the thickness of the medial OFC, considering a region related to value-guided decision making [33]. Bae et al. reported significantly thinner cortical thickness in patients with alcohol dependence than healthy comparison subjects in the left superior frontal cortical region [26]. In another study, Dai et al. demonstrated significant volumetric decreases in cortical thickness in the bilateral temporal, insular, precentral, and dorsolateral prefrontal gyri of patients with alcohol use disorder [27]. Likewise, cortical thickness alterations were supported by another investigation for both the left and right hemispheres [28]. These studies signify that cortical thickness deficits of the OFC might be related to alcohol or substance use disorder risk when OFC is considered to have an important role in controlling flexible, goal-directed behaviour, as well as its relationship with reward identification and acquisition. In this context, our present preliminary findings support this notion because, in the present study, both bilaterally reduced OFC volumes and bilaterally reduced OFC cortical thickness in patients with alcohol use disorder compared to those of healthy control subjects were found. It is considered that the OFC region seems to be closely related to impulsivity, which may be importantly associated with the behaviour of alcohol consumption.

Before concluding the discussion, it is important to highlight some limitations of the study. As findings should be accepted as preliminary data, these limitations should be taken into consideration when presenting projections for future studies on this subject. Firstly, unfortunately, the sample number was not as high as would be desired. Frankly, when the edition exclusion criteria were determined, it was expected that at least 30 patients would be identified during the inclusion period. This situation should be accepted as an important limitation. Secondly, this study focused on the OFC area and areas outside of the OFC were not taken into consideration. This situation can also be considered as a limitation. Thirdly, the manual tracing method was utilised when evaluating OFC volumes in this study. Despite the advantages of the method, it has some disadvantages as it requires direct individual care and attention. This is yet another limitation. On the other hand, the strengths of the present investigation included a good selection of cases in accordance with the exclusion criteria, meticulous administration of the scales, precision during the MRI scanning, and strict attention to comorbidity.

Consequently, it is suggested that the OFC region of the brain appears to be statistically significantly smaller in patients with alcohol use disorder, both in terms of cortical thickness and volume, compared to healthy controls. Future research should focus on the status of these relationships longitudinally and should assess the causality of the association with the treatment response.

## Figures and Tables

**Figure 1 brainsci-13-00552-f001:**
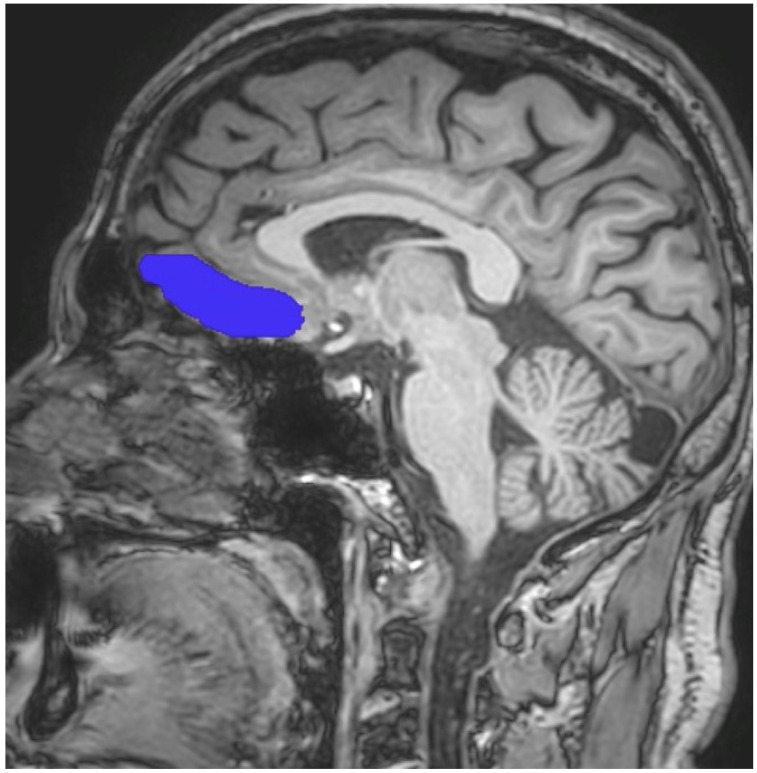
Sample manual tracing for the OFC.

**Figure 2 brainsci-13-00552-f002:**
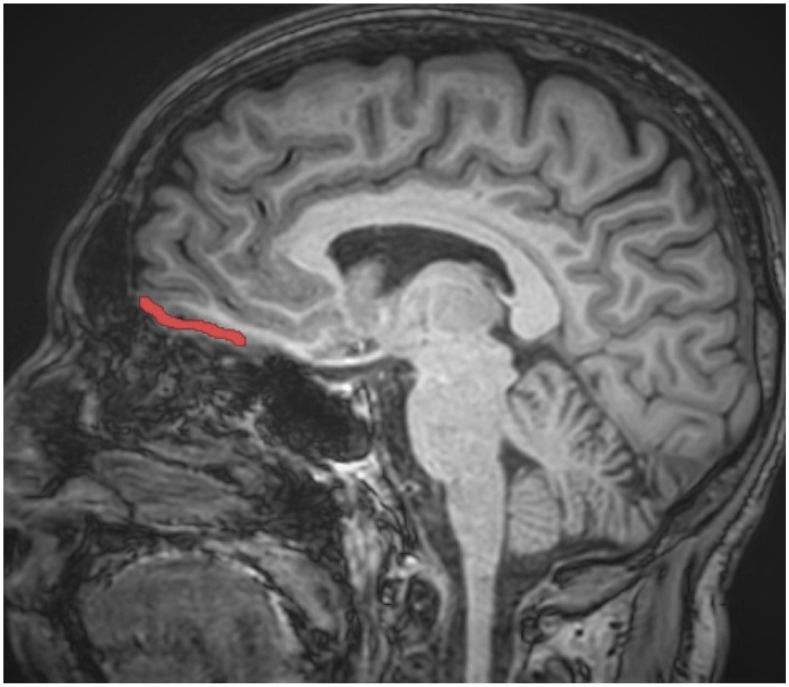
Sample cortical thickness measure for the OFC.

**Figure 3 brainsci-13-00552-f003:**
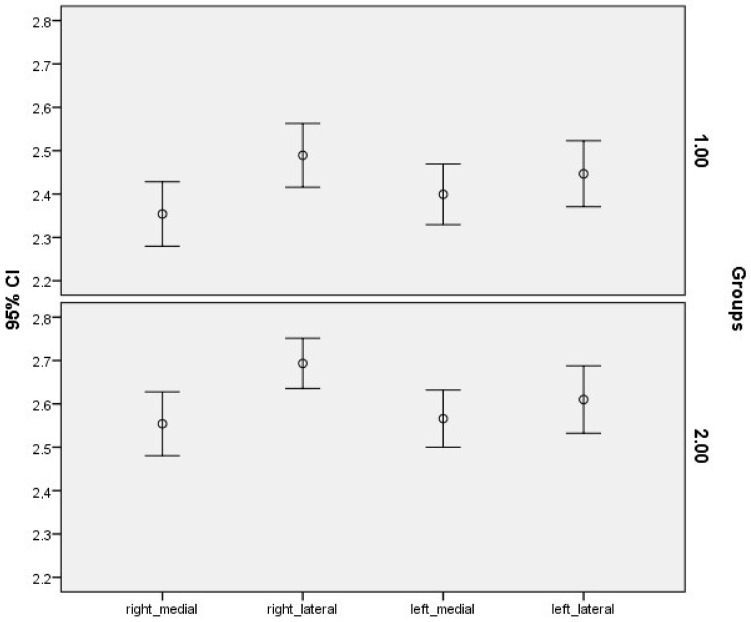
Error bars for cortical thickness in groups. 1: Patient group, 2: Controls.

**Figure 4 brainsci-13-00552-f004:**
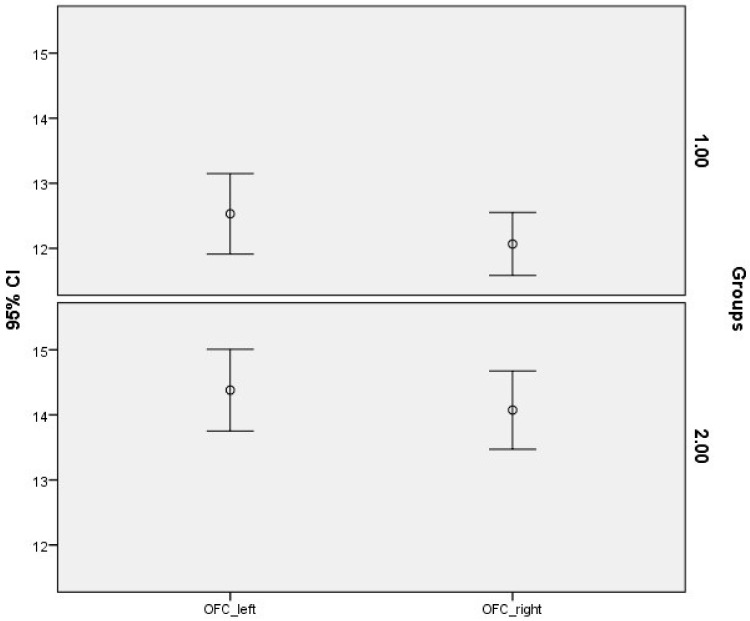
Error bars for OFC volumes in groups. 1: Patient group, 2: Controls.

**Table 1 brainsci-13-00552-t001:** Demographic, clinical, and orbitofrontal cortex volumes data in patients and the control group.

	Healthy Control Group (*n* = 17)	Patient Group (*n* = 15)	*p*
Age (years)	36.86 ± 11.03	32.52 ± 5.90	0.169
Handedness (right)	17	15	>0.05
Gender (female/male)	2/15	1/14	0.548
Length of illness (years)	-	16.60 ± 10.78	
HDRS scores	7.1 ± 1.9	11.7 ± 3.0	<0.001
MAST	-	28.40 ± 12.51	
SADQ-C	-	26.86 ± 14.83	
AUDIT	-	23.93 ± 7.46	
Grey matter volume	694.11 ± 44.13	668.00 ± 66.78	0.21
White matter volume	616.17 ± 29.44	619.00 ± 45.98	0.83
Orbitofrontal cortex volumes (cm^3^)			
Left	14.38 ± 1.13	12.53 ± 1.12	<0.001
Right	14.07 ± 1.09	12.07 ± 0.87	<0.001
Orbitofrontal cortex thickness (mm)			
Left medial	2.40 ± 0.13	2.57 ± 0.12	<0.01
Left lateral	1.46 ± 0.25	1.21 ± 0.16	<0.01
Right medial	2.35 ± 0.13	2.55 ± 0.13	<0.001
Right lateral	2.45 ± 0.14	2.61 ± 0.14	<0.001

Volumes presented are in cubic centimetres (cm^3^). MAST: Michigan Alcoholism Screening Test, SADQ-C: Severity of Alcohol Dependence Questionnaire, AUDIT: Alcohol Use Disorders Identification Test.

## Data Availability

The study did not report any data.

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
