# Peer review of "Cortical Thickness of the Orbitofrontal Cortex in Patients with Alcohol Use Disorder"

_brainsci, 2023, doi:10.3390/brainsci13040552_

Round 1

Reviewer 1 Report

The present study describes a reduced cortical thickness of the orbitofrontal cortex in patients with an alcohol use disorder. The finding of this study are in with earlier studies displaying reduced orbitofrontal cortex functioning in alcohol use, abuse and dependence. The study is sound, but more references should be made to the existing literature regarding AUD and orbitofrontal disfunction.

- See for example Moorman DE (2018) The role of the orbitofrontal cortex in alcohol use, abuse, and dependence. Prog Neuropsychopharmacol Biol Psychiatry. 2018 Dec 20; 87(Pt A): 85–107.

This study gives an overview of all the work on orbitofrontal cortex problems and AUD. For example, longitudinal work; the relationship between AUD and orbitofrontal functioning and recovery.

-in the present study, abstinence is not mentioned. Were patients sober at the time of testing? How long?

- The discussion directly repeats some of the findings in the results section (with numbers). It would be relevant to include the numbers in the results section and to review the findings in the discussion.

- In the limitations: why was the sample number lower than expected?

- Why was the audit missing for controls? Are we certain that they did not drink?

Author Response

It is attached

Reviewer 2 Report

This is an interesting study; however, the rationale of the study is not well justified since previous studies with larger samples have addressed similar questions (see for example: Dai et al., 2023; Momenan et al., 2012, but also Bae et al., 2016 for detoxified alcohol-dependent patients). Moreover, it should be mentioned that the authors do not report the aforementioned studies in their manuscript.

My major concern refers to the fact that the sample size consists of a significant barrier for the statistical analysis that the authors chose to conduct, such as t-test and GLM. Furthermore, results are not being reported appropriately. Error bar charts perhaps may provide significant information with regard to groups’ differences and variance. A table for GLM results should be included. 

Findings are not extensively discussed within the framework of current literature and more importantly the authors do not delve into the contribution of current study given the limited number of participants. 

Extensive editing of English language and style required, several parts of the text appear incomprehensible.

References:

Bae, S., Kang, I., Lee, B. C., Jeon, Y., Cho, H. B., Yoon, S., ... & Choi, I. G. (2016). Prefrontal cortical thickness deficit in detoxified alcohol-dependent patients. Experimental neurobiology, 25(6), 333.

Dai, X., Yu, J., Gao, L., Zhang, J., Li, Y., Du, B., ... & Zhang, H. (2023). Cortical thickness and intrinsic activity changes in middle-aged men with alcohol use disorder. Alcohol, 106, 15-21.

Momenan, R., Steckler, L. E., Saad, Z. S., van Rafelghem, S., Kerich, M. J., & Hommer, D. W. (2012). Effects of alcohol dependence on cortical thickness as determined by magnetic resonance imaging. Psychiatry Research: Neuroimaging, 204(2-3), 101-111.

Author Response

It is attached.

Round 2

Reviewer 1 Report

fine